# HantaNet: A New MicrobeTrace Application for Hantavirus Classification, Genomic Surveillance, Epidemiology and Outbreak Investigations

**DOI:** 10.3390/v15112208

**Published:** 2023-11-02

**Authors:** Roxana Cintron, Shannon L. M. Whitmer, Evan Moscoso, Ellsworth M. Campbell, Reagan Kelly, Emir Talundzic, Melissa Mobley, Kuo Wei Chiu, Elizabeth Shedroff, Anupama Shankar, Joel M. Montgomery, John D. Klena, William M. Switzer

**Affiliations:** 1Laboratory Branch, Division of HIV Prevention, Centers for Disease Control and Prevention, Atlanta, GA 30329, USAikb6@cdc.gov (A.S.); bis3@cdc.gov (W.M.S.); 2Viral Special Pathogens Branch, Division of High Consequence Pathogens and Pathology, Centers for Disease Control and Prevention, Atlanta, GA 30329, USAnkf2@cdc.gov (M.M.); ubu2@cdc.gov (E.S.); irc4@cdc.gov (J.D.K.); 3General Dynamics Information Technology, Atlanta, GA 30329, USA; qwu3@cdc.gov (E.M.); ylb9@cdc.gov (R.K.);

**Keywords:** MicrobeTrace, visualization, networks, hantavirus, outbreak, viral classification, bioinformatics, surveillance, epidemiology

## Abstract

Hantaviruses zoonotically infect humans worldwide with pathogenic consequences and are mainly spread by rodents that shed aerosolized virus particles in urine and feces. Bioinformatics methods for hantavirus diagnostics, genomic surveillance and epidemiology are currently lacking a comprehensive approach for data sharing, integration, visualization, analytics and reporting. With the possibility of hantavirus cases going undetected and spreading over international borders, a significant reporting delay can miss linked transmission events and impedes timely, targeted public health interventions. To overcome these challenges, we built HantaNet, a standalone visualization engine for hantavirus genomes that facilitates viral surveillance and classification for early outbreak detection and response. HantaNet is powered by MicrobeTrace, a browser-based multitool originally developed at the Centers for Disease Control and Prevention (CDC) to visualize HIV clusters and transmission networks. HantaNet integrates coding gene sequences and standardized metadata from hantavirus reference genomes into three separate gene modules for dashboard visualization of phylogenetic trees, viral strain clusters for classification, epidemiological networks and spatiotemporal analysis. We used 85 hantavirus reference datasets from GenBank to validate HantaNet as a classification and enhanced visualization tool, and as a public repository to download standardized sequence data and metadata for building analytic datasets. HantaNet is a model on how to deploy MicrobeTrace-specific tools to advance pathogen surveillance, epidemiology and public health globally.

## 1. Introduction

Hantaviruses are rodent-borne, negative-sense, tri-segmented RNA viruses that cause pulmonary and renal complications in infected humans [1,2,3]. Human infection typically occurs by contact with host reservoir excreta [2], but limited human-to-human transmission has been documented for the Andes virus (ANDV) species [4,5,6,7,8]. Hantaviruses are roughly organized into Old and New World designations based on viral genomics, geographic distribution and symptom manifestation [9,10,11,12,13]. Old World hantaviruses, like Seoul virus (SEOV), are typically distributed in Asia and Europe, with limited locations in the United States (US) and Canada, and cause hemorrhagic fever with renal syndrome (HFRS) [3,14,15,16]. Additional Old World hantaviruses causing HFRS include Hantaan, Dobrava-Belgrade, Puumala and Tula orthohantaviruses, which are distributed in Central/Eastern Europe, Asia and Europe [13,16,17]. New World hantaviruses cause hantavirus cardiopulmonary syndrome (HPS) and are predominantly found in the Americas [2,18,19], associated with Sin Nombre virus (SNV) species infections in the US [20,21,22,23], and ANDV in South America [7,24,25]. Across the Americas, additional New World hantaviruses causing human disease include Bayou, Black Creek Canal, Monongahela [26,27] and New York orthohantaviruses [28] in North America [11,13,17] and Choclo, Araraquara, Laguna Negra, Lechiguanas, Oran and Tunari orthohantaviruses in South America [11,17,29]. Additionally, hantaviruses have been discovered in Africa, but human infections have not been well documented yet [30].

From 1993 to 2021, 850 hantavirus cases were reported in the US [31]. These cases were laboratory-confirmed by the presence of hantavirus-reactive IgM, rising hantavirus-specific IgG titers, presence of hantavirus RNA, or hantavirus-reactive immunohistochemistry and symptoms consistent with hantavirus infection [32]. A few imported human cases of ANDV associated with travel to South America have also been reported in the US [4,32]. True HPS disease burden may be underestimated due to mild patient symptoms and unrecognized human and/or rodent host disease prevalence [2,19].

The tri-segmented hantavirus genome consists of the small (S), medium (M) and large (L) segments that encode nucleoprotein, glycoprotein and RNA polymerase, respectively [1,2]. Sequencing evidence from isolated hantavirus outbreaks demonstrates the utility of using hantaviral sequence information for molecular epidemiology. For example, Martinez et al. [6] used ANDV sequence data to demonstrate human-to-human ANDV transmission during a 2018–2019 outbreak in Argentina. Goodfellow et al. [33] and Kjemtrup et al. [23] demonstrated the geographic relatedness of human and rodent hantaviral sequences, and that molecular epidemiology can allow for predictive exposure risks as new hantaviral sequences are acquired. Centers for Disease Control and Prevention (CDC) collaborations with state public health laboratories and academia during hantavirus outbreaks have facilitated a broader picture of hantavirus diversity and spread [32,34,35]. However, there is a need for better coordination of nationwide hantavirus sampling in human and rodent hosts with real-time data integration, analytics and reporting tools to support surveillance, epidemiology and outbreak response. For instance, Kim et al. [36,37] highlighted the need for collaboration between health experts and bioinformaticians to improve hantavirus genomic epidemiology and surveillance, and suggested the use of advanced and accessible bioinformatic tools that allow for real-time data integration, visualization and analysis.

To fill this gap, we developed and validated a new MicrobeTrace-powered visualization tool called HantaNet, using published hantavirus sequence and contact tracing data [6,32,34]. MicrobeTrace is a secure web browser tool developed at the CDC [38] that remains operational without an internet connection and facilitates data integration and analysis with a suite of customizable views, including networks, phylogenetic trees, world maps, tables and charts that could be tailored as a dashboard view for real-time data analytics and reporting. MicrobeTrace has been widely used to integrate, visualize and analyze data from several pathogens, including HIV [39,40,41,42,43], hepatitis [44,45], tuberculosis [46] and SARS-CoV-2 [47,48,49]. We describe the expansion of MicrobeTrace as a pathogen-specific tool for genomic surveillance and variant classification. HantaNet is a free, secure and user-friendly web tool (https://cdcgov.github.io/HantaNet/ (accessed on 23 October 2023)) to integrate, visualize and analyze molecular and epidemiological data for rapid public health action.

## 2. Materials and Methods

### 2.1. Database Search and Standardization of Hantavirus Data for Use in HantaNet

We searched the National Center for Biotechnology Information (NCBI) GenBank database (https://www.ncbi.nlm.nih.gov/genbank/ (accessed on 28 June 2022)) for all hantavirus S, M and L nucleotide sequences collected prior to 2021 using an in-house Python script (https://github.com/evk3/hantavirus_US_distribution/blob/main/DL_Hanta_genomes.ipynb (accessed on 28 June 2022)). The S (1287 nucleotides (nt)), M (3423 nt) and L (6474 nt) coding sequences were aligned separately with the Multiple Alignment using Fast Fourier Transform (MAFFT) online tool [50,51] to generate FASTA alignments, which were manually inspected in BioEdit [52]. These preliminary FASTA alignments were loaded into MicrobeTrace to explore the data and estimate clustering genetic distance thresholds of Old and New World hantaviruses (the unreleased version is available at https://mossy426.github.io/HantaNet/ (accessed on 4 March 2022)). To build the hantavirus reference datasets, we searched the Bacterial and Viral Bioinformatics Resource Center (BV-BRC) database (https://www.bv-brc.org/ (accessed on 28 June 2022)) to retrieve curated S, M and L gene sequences associated with genomes representative of the US hantavirus strains. We loaded multiple sequence alignments with partial and full-length sequences to test their clustering in MicrobeTrace. This step was useful to perform a quality assessment of the reference alignments, and to exclude poor-quality and short-length (<400 nt) sequences from the reference datasets. We created a standardized and shareable CSV template file for hantaviral sequence and metadata submission to HantaNet (https://github.com/CDCgov/HantaNet/wiki/Sequence-and-Metadata-Submission-Form (accessed on 23 October 2023)) after data consolidation with Microsoft Excel and Tableau Desktop (version 2021.4.3) software tools.

### 2.2. Building Hantavirus Reference Gene Modules in MicrobeTrace for HantaNet

The open-source CDC MicrobeTrace GitHub repository (https://github.com/CDCgov/MicrobeTrace (accessed on 4 March 2022)) was cloned to create the HantaNet GitHub repository (https://github.com/CDCgov/HantaNet (accessed on 23 October 2023)). The three reference FASTA alignments and standardized metadata were initially loaded into three separate MicrobeTrace sessions to build the S, M and L visualization modules, respectively. These three modules were built using the Tamura-Nei 93 (TN93) nucleotide substitution model [53] implemented in MicrobeTrace to compute the pairwise genetic distances between the aligned sequences [54,55] and to empirically estimate the best genetic distance threshold for hantavirus strain clustering consistent with S, M and L gene segment phylogenies [10,32,56,57]. For example, if two different strains cluster together, the genetic distance cutoff can easily be decreased in HantaNet to remove the genetic links and separate the strains for determining the distance threshold. Once validated, the S, M and L modules were saved as three MicrobeTrace session files in the HantaNet GitHub repository where the application is hosted. 

### 2.3. Validation of the HantaNet Tool

Any modern web browser can be used to open the HantaNet tool (https://cdcgov.github.io/HantaNet/ (accessed on 23 October 2023)) after accepting the end-user license agreement. Depending on the gene segment to be analyzed, the user first selects the pre-loaded reference module of interest to launch the default 2D network analysis view (see Video S1 published at https://www.youtube.com/watch?v=7j2KZ4tLStU (accessed on 23 October 2023) for a demonstration of the HantaNet tool). New hantavirus test sequences and metadata from US genomic surveillance provided by Whitmer et al. [32] were used to validate the HantaNet tool and confirm hantavirus clustering and classification with the reference dataset. Briefly, we downloaded the S, M and L reference datasets from HantaNet and aligned the new sequences to each reference alignment using the “Add fragmentary sequence(s) to existing alignment” option in MAFFT, which aligns the new sequences against the references, while keeping the reference alignment intact (refer to our protocol at https://github.com/CDCgov/HantaNet/wiki/Sequence-Alignment-Protocols (accessed on 4 October 2022)). This step ensured that the sequence alignments were standardized before loading into HantaNet for reproducible distance calculations and comparable results. We also added the new metadata to the standardized template, as described above. The FASTA alignments and metadata were loaded into HantaNet, and the specific genetic distance threshold previously determined for each gene segment was set up to cluster the sequences by hantavirus strain in the 2D network analysis view. The genetic relatedness and clustering of the hantavirus sequences in the 2D networks at the identified distance thresholds were confirmed in the phylogenetic tree view in HantaNet [58] by comparison with published hantavirus phylogenies [10,32,56,57]. We also built phylogenetic trees using the neighbor-joining method in MEGA X [59,60] and loaded the Newick tree output into HantaNet, which calculates the patristic distances from the Newick tree branch lengths to generate the network [61,62]. The new genetic links from the calculated patristic distances were visualized in HantaNet using the 2D network and the phylogenetic tree views.

We further validated HantaNet for building transmission networks by loading published sequence and contact tracing data from two different outbreak investigations [6,34]. In order to build and analyze the transmission networks in HantaNet, we applied the nearest neighbor (NN) algorithm to construct the minimum spanning tree (MST), which determines the shortest genetic distance path through the network [63,64]. The link width in the NN network was adjusted to be proportional to the genetic distance to easily visualize the genetic differences between sequences. The networks generated in HantaNet were compared to the published networks [6,34].

### 2.4. HantaNet Improvements and New Features

We built functionality in HantaNet to add new icons in the networks to represent the origin of sequences as a human, animal or viral isolate. These options are customizable, and more icons can be added per user preferences (please refer to our protocol at https://github.com/CDCgov/HantaNet#adding-custom-symbols-png-jpg-and-svg-to-be-used-in-the-github-web-application---collaborators-only (accessed on 4 October 2022)). We also added a new directional arrow feature in the network view to allow users to further characterize potential transmission directionality. This is an optional feature that can be turned on or off for unidirectional and bidirectional arrows to visualize the order of transmission events. We modified polygon labels to keep the text font case as entered in the metadata, which is useful for proper display of strain acronyms (e.g., SNV, SEOV) and geographic fields (e.g., country and state abbreviations). More view tabs can be added to the dashboards and are customizable. We also modified MST calculations for improved NN network display. We demonstrate how users can easily built interactive dashboards in HantaNet by integrating genetic, temporal and geographic data. The NN network can also be overlaid with map views and a timeline for spatiotemporal analysis.

## 3. Results

### 3.1. Validation of HantaNet for Classification of Hantaviruses

We built the HantaNet web tool by adapting MicrobeTrace to integrate curated S, M and L nucleotide sequence alignments and standardized metadata from US hantaviruses isolated from human and animal hosts (1984–2016). Table 1 contains the list of the 41 hantavirus genomes used to build the three HantaNet gene segment modules. The hantavirus genome sequences are publicly available in the GenBank database and the curated reference alignments can be downloaded from HantaNet (Figure 1).

The TN93 genetic distance threshold for clustering hantavirus species was empirically determined for each of the modules using known strain phylogenetic clustering and ranged from 12 to 20% genetic divergence (0.120–0.200 nucleotide substitutions per site) based on the loaded S, M and L reference alignments. The default TN93 distance cutoff selected in HantaNet was 0.127 (12.7%) for the S segment, 0.200 (20%) for the M segment and 0.144 (14.4%) for the L segment modules in HantaNet. Phylogenetic trees were also built in HantaNet and compared to published hantavirus phylogenies [10,32,56,57] to validate the clustering and genetic distances of 12 hantavirus species in each of the modules. The final S module (Appendix A) had representative sequences of the 12 hantavirus strains (ANDV, *n* = 2; BAYV, *n* = 3; BCCV, *n* = 2; BRV, *n* = 2; ELMCV, *n* = 2; ILV, *n* = 3; MGLV, *n* = 2; MULV, *n* = 2; NYV, *n* = 3; PHV, *n* = 4; SEOV, *n* = 19; SNV, *n* = 23). The M module (Figure 2) included 10 strains (ANDV, *n* = 2; BAYV, *n* = 2; BCCV, *n* = 1; BRV, *n* = 2; ELMCV, *n* = 2; MGLV, *n* = 1; NYV, *n* = 4; PHV, *n* = 2; SEOV, *n* = 22; SNV, *n* = 28), and the L module (Appendix A) had 8 strains (ANDV, *n* = 2; BAYV, *n* = 2; BCCV, *n* = 1; MGLV, *n* = 1; NYV, *n* = 1; PHV, *n* = 2; SEOV, *n* = 18; SNV, *n* = 16). The S, M and L networks were saved as separate MicrobeTrace sessions (Figure 1) in the cloned MicrobeTrace GitHub repository built for HantaNet (https://github.com/CDCgov/HantaNet (accessed on 23 October 2023)).

### 3.2. Genomic and Epidemiology Analyses of Hantaviruses Using HantaNet

We validated HantaNet using data available from three hantavirus studies [6,32,34].

Case Study 1. HantaNet was validated with 21 genome sequences of hantaviruses isolated from human hosts during US genomic surveillance (1999–2021). At the 0.2 genetic distance cutoff (Figure 2), the clustering of the 21 new M gene sequences (yellow nodes) in the 2D network revealed that they are SNV (*n* = 18), MGLV (*n* = 2) and NYV (*n* = 1) hantavirus strains. When the genetic distance cutoff was decreased to 0.127 in HantaNet (Figure 3A), two SNV subgroups were observed, SNV1 (*n* = 3) and SNV2 (*n* = 8). Our phylogenetic analysis supported the network clustering in HantaNet (Figure 2 and Appendix A), and our results were consistent with the hantavirus phylogenetic clustering reported by Whitmer et al. [32]. Genetic distance and clustering differences were also observed among the S, M and L gene segments, as discussed by Whitmer et al. [32].

We performed spatiotemporal analysis to show the geographic location of the new M sequences compared to the reference dataset. First, we applied the NN algorithm, which finds the shortest genetic distance route in a network, reducing the inferred genetic links from 426 to 52 (Figure 3A). We retrieved the calculated genetic distances of the closest SNV sequences from the table view in HantaNet (Table 2) to confirm their SNV classification, as reported by Whitmer et al. [32]. To incorporate geographic data, the MST was overlaid on the map, using the map view in HantaNet to visualize the geographic distribution of SNV1 and SNV2 variants (Figure 3B).

Case Study 2. HantaNet was used to plot the genetic and contact tracing data from the 2018-2019 ANDV outbreak among 34 persons in Argentina reported by Martinez et al. [6], and the transmission network was inferred (Figure 4). The first reported case of the ANDV outbreak occurred in early November 2018 in Chubut Province, Argentina. The HantaNet transmission network reveals three major transmission events, as previously described [6] and represented by large human icons (ARG/1, ARG/2, ARG/9). According to the report, the first transmission event in November 2018 occurred when Patient 1 (ARG/1) was symptomatic and attended a party, where five other persons became infected (ARG/2-ARG/6). Patient 2 (ARG/2) was the spouse of Patient 9 (ARG/9), who also became infected together with five other persons (ARG/7-ARG/8, ARG/11-ARG/13) during the second transmission event in December 2018. The third transmission event in January 2019 occurred after Patient 2 died (ARG/2). Ten additional persons (ARG/15-ARG/19, ARG/22-ARG/24, ARG27, ARG/30) became infected while attending the wake of Patient 2 (ARG/2) and being exposed to Patient 9 (ARG/9), who was symptomatic.

Case Study 3. The transmission network for the 2017 SEOV outbreak at US and Canada facilities (home-based ratteries, local pet stores and homes) by Knust et al. [34] was inferred using HantaNet (Figure 5). This outbreak involved 31 facilities in 11 states and 2 confirmed facilities in Canada. Three major clusters of confirmed SEOV cases were observed in Illinois (*n* = 12), Wisconsin (*n* = 5) and Utah (*n* = 3), as shown in the inferred transmission network (Figure 5). Pet rats were commonly exchanged between US facilities, and confirmed rat movement is shown with black directional arrows. Twenty-eight facilities confirmed infected rats, and three facilities were associated with human infections (facilities 26, 30, 31). The collected viral sequences were highly similar to SEOV isolated from the English Cherwell clade, suggesting a possible importation to the US and Canada from the UK or Europe [34]. The network shows likely rat movement between Canadian facilities (nodes A and B) and several US facilities (nodes 9, 10, 12, 17 and 28), indicating potential transmission events. Unconfirmed rat movement is shown with orange directional arrows.

## 4. Discussion

MicrobeTrace was originally developed as a browser-based web application for secure, local HIV surveillance data analysis for cluster detection and outbreak response [40]. Its use has expanded for the study of other pathogen outbreaks [38,46,47,48,49]. During the 2022–2023 global mpox outbreak, the MicrobeTrace team at the CDC built a local Nextstrain [65] and MicrobeTrace pipeline for continued US mpox genomic surveillance and to support secure cluster analysis of HIV–mpox coinfections [66]. The browser-based compatibility of MicrobeTrace offers the advantage of analyzing sensitive data locally without transmitting information across the internet. We developed HantaNet (https://cdcgov.github.io/HantaNet/ (accessed on 23 October 2023)) as a use case for the rapid deployment of MicrobeTrace as a free and secure, pathogen-specific platform for data integration, analysis, visualization and data sharing. We show that MicrobeTrace is easily adaptable to study specific pathogens and new outbreaks and offers the flexibility to add new features and tool modifications based on user preferences. We validated HantaNet as a hantavirus classification and genomic epidemiology tool using genomic surveillance sequences provided by Whitmer et al. [32] and published sequence and contact tracing data [6,34] to create different visualizations, including networks, phylogenetic trees, maps and dashboards. During this process, we developed enhancements to HantaNet that have since been deployed in MicrobeTrace. For outbreak investigations, users can empirically determine the genetic distance threshold used for detection of transmission chains and clusters for a specific hantavirus involved in an outbreak by using a slider bar in the settings menu. HantaNet improves transmission network visualizations by the addition of icons and directional arrows. Like MicrobeTrace, HantaNet displays individual genetic and epidemiologic links as colored solid lines. Moreover, dashed lines represent links supported by both genetic and epidemiologic data, and the new directional arrow feature in the 2D network view shows the order of transmission events, which simplifies and improves the network visualization and comprehension of transmission during an investigation.

We validated the genetic relatedness for classification of hantaviruses in HantaNet by building genetic networks and phylogenetic trees, which were consistent with published hantavirus gene phylogenies [10,32,56,57]. While phylogenetic trees and networks are two different visualizations of the same data, transmission networks and clustering are generally both easier to understand and to explore than trees because network diagrams are intuitive, and the distance cutoff in HantaNet can easily be changed within a session. For example, we showed that hantaviral strain clusters are resolved with different S, M and L genetic distance thresholds. Genetic networks are also useful to simplify visualization of gene reassortment [67,68,69,70], which is a mechanism for the evolution of viruses with segmented genomes [71,72,73,74]. While visualization of gene reassortment is mostly performed with phylogenetic and hybrid plot methods [75,76], these tools require programming skills and are more time-consuming and less efficient when compared to genetic network building tools like MicrobeTrace [38]. New MicrobeTrace-powered tools, like HantaNet, could be deployed for the analysis and visualization of segmented genomes for comparison with existing methods to detect gene reassortment in other pathogens.

We showed that building and visualizing networks in HantaNet is a fast and reliable method for the study of contact tracing and sequencing data from outbreak investigations. We also demonstrated that HantaNet could be used for hantavirus genomic surveillance with the rapid creation of dashboards useful to further explore the spatiotemporal relationships of strain variants and potential human exposure hotspots based on viral sequence similarity. HantaNet and other MicrobeTrace-related applications could be modified for phylogeographic studies by integrating Markov chain (discrete or continuous trait models) and coalescent models for ancestral reconstruction [77], commonly used in the Bayesian Evolutionary Analysis Sampling Trees (BEAST) method [78,79,80] and Nextstrain platform [65,81,82] for phylogenetic inference. Bayesian inference methods can be time-consuming, but more efficient approaches to build time-scaled phylogenies like TreeTime are available as open-source code [83] and could be integrated into HantaNet and MicrobeTrace. Phylogeography could be useful to study hantavirus outbreaks and report changes that occur in host–virus distribution during genomic surveillance, including county-level data to determine if hantavirus strains are confined by state and national boundaries.

We designed HantaNet to be a single, unified platform for hantavirus data standardization, integration, visualization, analytics and sharing. While there are limited hantavirus sequences currently available in public databases like GenBank, we envision that HantaNet will provide a framework for rapid hantavirus data submission to improve data sharing and the availability of sequences in public repositories. HantaNet could also serve as a repository for rapid sequence and metadata sharing among the CDC, state public health laboratories and academic partners. To encourage timely hantaviral data analysis and sharing, the HantaNet GitHub project page provides instructions on how to use the tool (https://github.com/CDCgov/HantaNet/wiki (accessed on 23 October 2023)) and to submit new sequence and metadata to HantaNet using the standardized submission form (https://github.com/CDCgov/HantaNet/wiki/Sequence-and-Metadata-Submission-Form (accessed on 23 October 2023)). Also, users interested in adding new reference alignments to include other hantaviral strains can fork the current HantaNet GitHub repository to replace the S, M and L MicrobeTrace session files with the new FASTA alignments and standardized metadata. HantaNet users will benefit from using these resources to easily download standardized hantavirus metadata and sequence alignments to build their datasets for rapid data visualization, analysis and reporting during outbreak investigations.

In summary, HantaNet is a model on how to develop secure and freely accessible MicrobeTrace-powered tools to improve public health investigations. These pathogen-specific tools could also be deployed in less developed countries to include the study of neglected diseases. Deployment and use of MicrobeTrace applications will help overcome socioeconomical and geographical barriers to the advancement of public health infrastructure worldwide.

## Figures and Tables

**Figure 1 viruses-15-02208-f001:**
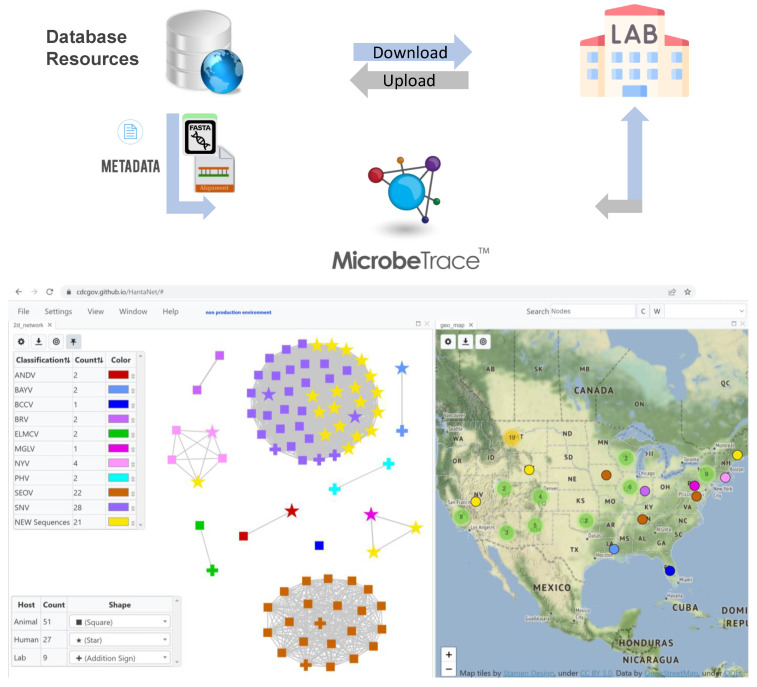
Building HantaNet reference modules for the small (S), medium (M) and large (L) gene segments. Sequence alignments and standardized metadata were integrated from a collection of hantavirus genomes available from GenBank and curated at the Bacterial and Viral Bioinformatics Resource Center (BV-BRC). The hantavirus S, M and L nucleotide sequences were aligned as described in the materials section. The reference alignments and metadata files were loaded and integrated as separate gene segment modules into MicrobeTrace to build the HantaNet tool for hantavirus classification, epidemiology and outbreak detection. Users can download the hantavirus reference datasets and use them to standardize their own laboratory and epidemiological data for visualization and analysis in HantaNet.

**Figure 2 viruses-15-02208-f002:**
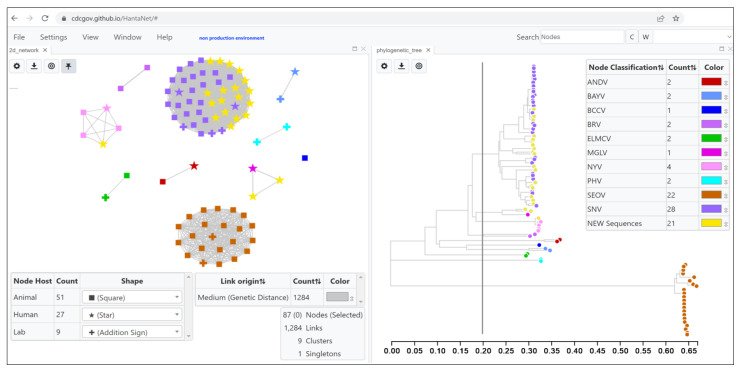
Example of a HantaNet dashboard built with the 2D network and unrooted phylogenetic tree views in the medium (M, glycoprotein) reference gene segment module. After loading the curated US hantavirus genomic surveillance data into HantaNet, the TN93 nucleotide substitution model was applied to generate a 2D network (**left**). The vertical grey line in the phylogenetic tree view (**right**) represents the 0.2 substitutions per site genetic distance cutoff used to obtain hantavirus strain-specific clusters (*n* = 10; nine clusters of two or more sequences and one singleton). Nodes represent individual sequences and were colored based on hantavirus classification, as shown in the key table next to the phylogenetic tree (**right**) (see Table 1 for abbreviations). Nodes were shaped by host: animal (square), human (star) and lab or viral isolate (addition sign).

**Figure 3 viruses-15-02208-f003:**
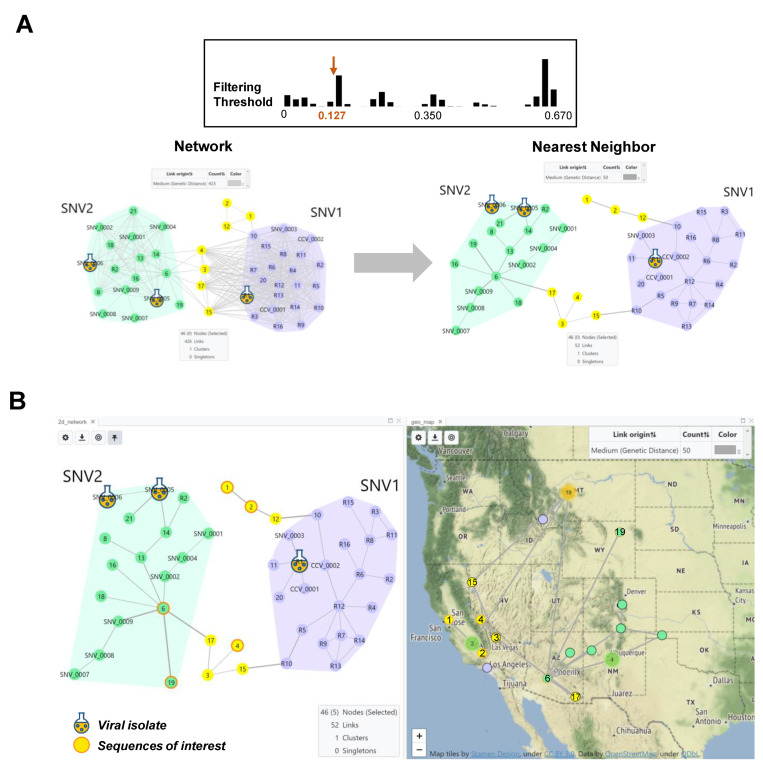
HantaNet can connect genetic networks into minimum spanning trees (MSTs) and geographic locations. (**A**) Comparison of the 2D network view generated with TN93 at a threshold of 0.127 nucleotide substitutions per site (**top**,**left**) and after applying the nearest neighbor (NN) algorithm (**right**). We analyzed 21 M gene segments of SNV collected in the US from 1999 to 2021. Two main groups were observed, SNV1 (purple polygon) and SNV2 (green polygon), consistent with findings from Whitmer et al. [32]. Reference SNV1 and SNV2 sequence nodes are labeled with the prefix R, SNV or CCV in the network. The yellow nodes represent new sequences that did not cluster with SNV1 or SNV2 and may represent novel strains. The NN algorithm was applied to find the MST or shortest genetic distance path in the network, which prunes more distant links between nodes, while retaining the connection between the two closest nodes. The genetic link width was pre-set to be proportional to the TN93 distance. (**B**) A dashboard built with the NN network (**left**) and map (**right**) views. The NN network was overlayed on the map to visualize virus geographic location and spread. The selected yellow nodes with orange borders (sequences 1, 2 and 4) represent examples of new sequences of interest (e.g., outbreak, genomic surveillance, new variants, etc.) to show their distribution in the map. Two reference SNV2 sequences (6 and 19) were also selected and displayed in the map (green nodes with orange borders). Individual sequences with associated state- and county-level metadata are shown in the map as small nodes with black borders. Three clusters of sequences were observed in the states of California (*n* = 2), New Mexico (*n* = 4) and Montana (*n* = 18), and are represented by the green or orange circles, respectively.

**Figure 4 viruses-15-02208-f004:**
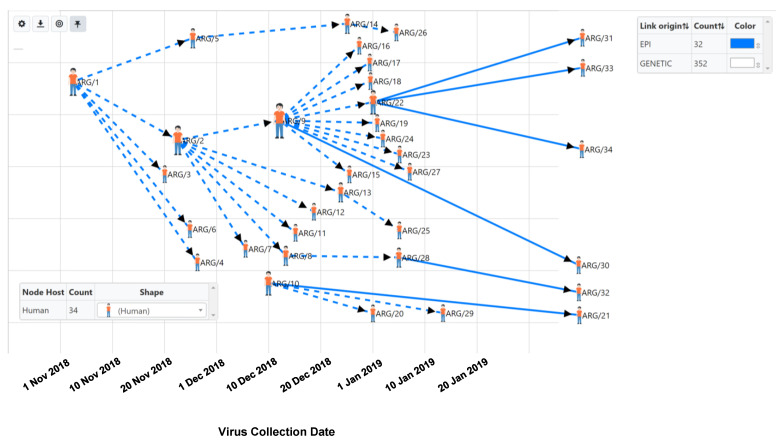
Transmission network of the 2018–2019 Andes virus (ANDV) outbreak in Argentina using available S gene sequences and contact tracing data [6] inferred with HantaNet, using a genetic distance cutoff of zero nucleotide substitutions per site (352 genetic links are hidden). The nodes were customized with human icons to represent patients and sized by the number of secondary transmissions. Blue lines show epidemiologic links, which display directional arrows where confirmed transmission events occurred. Dashed lines show both genetic and epidemiologic linkage providing stronger support for the transmission event. The index case was patient ARG/1, shown on the upper left of the network. Virus collection dates are shown with gray vertical lines in the network layout.

**Figure 5 viruses-15-02208-f005:**
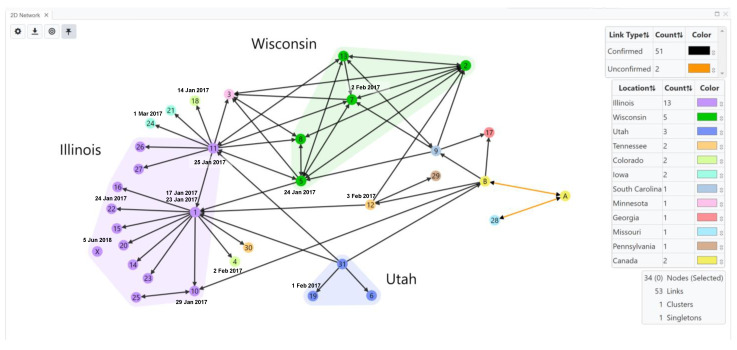
Transmission network of Seoul virus (SEOV)-infected rats in home-based ratteries, local pet stores and homes during the 2017 multistate outbreak in the United States (US) and Canada [34]. Public SEOV sequences and confirmed facility data (laboratory-confirmed SEOV infections in people or rats) were loaded into HantaNet to build the transmission network visualization. Nodes were colored by location, represented by circles with facility numbers (US, *n* = 31) or letters (Canada, *n* = 2). Polygons represent clusters of facilities in US states with the greatest number of confirmed cases (Illinois, purple; Wisconsin, green; Utah, blue). Nodes were labeled with the sample collection dates (M/D/YY) and represent SEOV sequence availability. Links show unidirectional or bidirectional arrows, representing confirmed (black links) or unconfirmed (orange links) movement of rats between facilities. A total of 17 human cases were identified at the following US facilities: 26 (purple node), 30 (orange node), and 31 (blue node) in Illinois, Tennessee and Utah, respectively. Node X (purple circle) shows that a facility in Illinois had an infected pet rat detected after the outbreak in June 2018.

**Table 1 viruses-15-02208-t001:** Hantavirus reference sequences used to build the HantaNet gene segment modules.

Classification ^a^	Virus Name ^b^	Virus ID ^c^	Collection Date ^d^	Host ^e^	Isolation Country ^f^	State ^g^	Segment ^h^	Accession ID ^i^
NDV	Andes virus	ANDV_0001	1997	Rice rat	Chile	Unknown	S M L	AF291702 AF291703 AF291704
		ANDV_0002	2016	Human	Switzerland	Unknown	S M L	KY659432 KY604962 KY659431
BAYV	Bayou virus	BAYV_0001	1993	Human	United States of America	Louisiana	S M	L36929 L36930
		BAYV_0002		Viral isolate			S M L	GQ200820 GQ244521 GQ244526
		BAYV_0003		Viral isolate			S L	NC_038298 NC_038299
BCCV	Black Creek Canal virus	BCCV_0001	1994	Cotton rat	United States of America	Florida	S M	L39949 L39950
		BCCV_0002		Viral isolate			S	NC_043075
BRV	Blue River virus	BRV_0001	2005	Deer mice	United States of America	Missouri	S	DQ090888
		BRV_0002	2005	Deer mice	United States of America	Missouri	S	DQ090890
		BRV_0003	1997	Deer mice	United States of America	Indiana	M	AF030551
		BRV_0004	1997	Deer mice	United States of America	Oklahoma	M	AF030552
ELMCV	El Moro Canyon virus	ELMCV_0001	1994	Harvest mice	United States of America	California	S M	U11427 U26828
		ELMCV_0002		Viral isolate			S M	NC_038423 NC_038424
ILV	Isla Vista virus	ILV_0001	1994	Vole	United States of America	California	S	U19302
		ILV_0002	1995	Vole	United States of America	California	S	U31534
		ILV_0003	1995	Deer mice	United States of America	California	S	U31535
MGLV	Monongahela virus	MGLV_0001	1985	Deer mice	United States of America	West Virginia	S	U32591
		MGLV_0002	1997	Human	United States of America	Pennsylvania	S M L	MH539867 MH539866 MH539865
MULV	Muleshoe virus	MULV_0001	1995	Cotton rat	United States of America	Texas	S	U54575
		MULV_0002	1999	Cotton rat	United States of America	Texas	S	KX066124
NYV	New York virus	NYV_0001	1994	Deer mice	United States of America	New York	S M L	MG717391 MG717392 MG717393
		NYV_0002	1994	Human	United States of America	Rhode Island	S M	U09488 U36801
PHV	Prospect Hill virus	PHV_0001		Viral isolate			S M L	NC_038938 NC_038940 NC_038939
SEOV	Seoul virus	SEOV_0001	1993	Rat	South Korea	Unknown	S M L	NC_005236 S47716 X56492
		SEOV_0002	2013	Rat	United States of America	New York	S M	KJ950866 KJ950862
		SEOV_0003	2002	Rat	United States of America	Maryland	S M L	KT897726 KT897725 KT897724
		SEOV_0004		Viral isolate			S M L	KU204960 KU204959 KU204958
		SEOV_0005	2013	Rat	United States of America	New York	S M	KJ950867 KJ950863
		SEOV_0006	2013	Rat	United States of America	New York	S M	KJ950868 KJ950864
		SEOV_0007	2013	Rat	United States of America	New York	S M	KJ950869 KJ950865
SNV	Sin Nombre virus	SNV_0001	1993	Human	United States of America	New Mexico	S M L	NC_005216 NC_005215 L37901
		SNV_0002	1993	Deer mice	United States of America	New Mexico	S M L	L37904 L37903 L37902
		SNV_0003	2009	Deer mice	United States of America	Montana	S M	JQ690281 JQ690283
		SNV_0004	1993	Human	United States of America	New Mexico	S M	L25784 L25783
		SNV_0005		Viral isolate			S M L	KF537003 KF537002 KF537001
		SNV_0006		Viral isolate			S M L	KF537006 KF537005 KF537004
		SNV_0007	2008	Deer mice	United States of America	Montana	S M	JQ690276 JQ690279
		SNV_0008	2008	Deer mice	United States of America	Montana	S M	JQ690277 JQ690280
		SNV_0009	2009	Deer mice	United States of America	Montana	S M	JQ690282 JQ690284
	Convict Creek virus	CCV_0001	1993	Deer mice	United States of America	California	S M L	L33683 L33474 AF425256
		CCV_0002	1995	Deer mice	United States of America	California	S M	L33816 L33684

^a^ Hantavirus strain abbreviation. ^b^ Virus strain name entered as standard full name. ^c^ Unique virus or specimen identifier assigned manually. ^d^ Collection dates of original viruses are not included for viral isolates. ^e^ Common name of the natural host of the organism from which the sample was obtained. Viral isolates include viruses propagated in cell culture or laboratory animals, and NCBI curated non-redundant sequences (RefSeq). ^f^ Country where virus specimen was collected. Country is not included for viral isolates. ^g^ State or province where virus specimen was collected. State is not included for viral isolates. ^h^ Refers to Small (S), Medium (M) or Large (L) gene segments. ^i^ GenBank unique gene identifier.

**Table 2 viruses-15-02208-t002:** Sin Nombre virus (SNV) M segment pairwise genetic distances (substitutions/site) calculated using the TN93 nucleotide substitution model.

SNV Group	Pairwise Genetic Distance
SNV1	0–0.05714
SNV1-like	0.07099–0.11967
SNV2	0.00029–0.06914
SNV2-like	0.12026–0.12295
Other	0.00693–0.03564

## Data Availability

The sequence data used to build the HantaNet modules are publicly available in the GenBank database and can be downloaded at https://cdcgov.github.io/HantaNet/ (accessed on 23 October 2023). Additional data presented in this study are available upon request from the corresponding authors.

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
