# Peer review of "HantaNet: A New MicrobeTrace Application for Hantavirus Classification, Genomic Surveillance, Epidemiology and Outbreak Investigations"

_viruses, 2023, doi:10.3390/v15112208_

Round 1
Reviewer 1 Report
Comments and Suggestions for Authors
Hantaviruses zoonotically infect humans worldwide causing hantavirus cardiopulmonary syndrome in Americas and hemorrhagic fever with renal syndrome predominantly in Europe and Asia. Genetic evidence suggests that viral sub-strains are geographically confined, but the lack of an organized clearing house for hantavirus data integration and sharing hinders virus surveillance and outbreak response. To overcome this challenge, authors have developed HantaNet, a MicrobeTrace based application for rapid classification of hantaviruses. CDC and public health laboratories already use MicrobeTrace for HIV, COVID-19 and monkeypox cluster detection and response. The HantaNet web tool was developed by adapting MicrobeTrace to integrate sequence alignments and standardized metadata from US hantaviruses isolated from human and animal hosts. The reference alignments and metadata files were imported as three separate gene segment modules, for the S, M, and L segment of hantavirus genome, into the open-source CDC MicrobeTrace repository to create HantaNet GitHub repository. The S, M, and L modules were built using the Tamura-Nei 93 (TN93) nucleotide substitution model to calculate the pairwise genetic distances between the aligned sequences and to empirically estimate the best genetic distance threshold for hantavirus strain clustering. To build and analyse the transmission networks in HantaNet, the nearest neighbor (NN) algorithm to construct the minimum spanning tree which determines the shortest genetic distance path through the network was used. The NN network can also be overlaid with map views and timeline for spatiotemporal analysis. To validate the HantaNet tool and confirm hantavirus classification, data available from three hantavirus studies were used (Whitmer et al., Martinez et al. and Knust et al.). The HanaNet platform enables user to upload, subtype and classify hantavirus sequences, to visualize the data in a network or phylogenetic tree, to overlay molecular and epidemiological links and to visualize spatiotemporal distribution of hantaviruses.
HantaNet is a unified Web platform for hantavirus data submission, analysis, classification, and outbreak detection making it a versatile tool for rapid deployment during surveillance and outbreak response and therefore of great relevance to the field. The manuscript is written in clear and comprehensive manner. The statement and conclusions drown are consistent with the evidence and arguments presented and are supported by listed citations. The tables and figures in the manuscript and supplementary data are appropriate and easy to interpret and understand. Therefore, I highly recommend manuscript for the publication and have only a few minor coments/questions:
General comments, questions:
1. In which file formats novel viral sequences should be uploaded to the HantaNet? Can new viral sequences simply be loaded into HantaNet or multiple sequence alignment tool must be used first to align the new sequences against the aligned reference?
2. Can maybe a phylogenetic tree can be uploaded to HantaNet?
3. Are any sequence quality control and source control incorporated for the viral sequences uploaded to the HantaNet? Does HantaNet automatically update as new sequences from public databases and repositories are uploaded?
3. Can the HantaNet be used for of Hantaviruses of the Old World to any extent? Or the new application with Old World Hantavirus sequences developed?
Specific comments:
Line 11: Rephrase “are transmitted by infected rodents”, rodents are host and source of hantaviruses, infection is transmitted via aerosol.

Reviewer 2 Report
Comments and Suggestions for Authors
The manuscript titled "HantaNet: A New MicrobeTrace Application for Hantavirus Classification, Genomic Surveillance, Epidemiology and Outbreak Investigations" by Cintron et al is a very interesting piece of work reported here. The authors developed a standalone visualization engine, HantaNet, for hantavirus genomes that would facilitates viral surveillance and classification for early hantavirus outbreak detection. This tool can be used without access to the internet. This would be useful for timely analysis of hantavirus genomes and aid to the understanding of the hantavirus epidemiology. The research presented here holds high significance and novelty.
I thank the authors for sharing the informative video presentation explaining the functioning of the developed tool. The research presented here is very well designed and the manuscript is well written. I would only suggest the authors:
1. to add brief information, in the first Introductory paragraph, on various species of hantaviruses reported worldwide.
2. Line 327: 2022-23 global mpox outbreak.
Well done!
